



# ICE SHEET AND SEA ICE ULTRAWIDEBAND MICROWAVE RADIOMETRIC AIRBORNE EXPERIMENT (ISSIUMAX) IN ANTARCTICA: FIRST RESULTS FROM TERRA NOVA BAY

Marco Brogioni[1], Mark J. Andrews[2], Stefano Urbini[3], Kenneth C. Jezek[4], Joel T. Johnson[2], Marion Leduc-Leballeur[1], Giovanni Macelloni[1], Stephen F. Ackley[5], Alexandra Bringer[2], Ludovic Brucker[6], Oguz Demir[2], Giacomo Fontanelli[1], Caglar Yardim[2], Lars Kaleschke[7], Francesco Montomoli[1], Leung Tsang[8], Silvia Becagli[9], and Massimo Frezzotti[10]

[1]"N.Carrara" Institute of Applied Physics - National Reasearch Council, IFAC-CNR, Sesto Fiorentino, 50019, Italy
[2]Electroscience Laboratory, The Ohio State University, Columbus, OH 43212, USA
[3]Istituto Nazionale di Geofisica e Vulcanologia - INGV, Rome, 00143, Italy
[4]Byrd Polar and Climate Research Center, The Ohio State University, Colombus, OH 43210, USA
[5]Department of Earth and Planetary Sciences, University of Texas at San Antonio, San Antonio, TX 78249, USA
[6]Center for Satellite Application and Research NOAA/NESDIS and the U.S. National Ice Center, College Park, MD 20740, USA
[7]Alfred-Wegener-Institut, Helmholtz-Zentrum für Polar- und Meeresforschung, Bremerhaven, Germany
[8]Radiation Laboratory, University of Michigan, Ann Arbor, MI 48109-2122, USA
[9]"U. Schiff" Department of Chemistry, University of Florence, Sesto Fiorentino, 50019, Italy
[10]Department of Science, Università degli Studi Roma Tre, Rome, 00154, Italy

**Correspondence:** Marco Brogioni (m.brogioni@ifac.cnr.it)

**Abstract.** An airborne microwave wide-band radiometer (500-2000 MHz) was operated for the first time in Antarctica to better understand the emission properties of sea ice, outlet glaciers and the interior ice sheet from Terra Nova Bay to Dome C. The different glaciological regimes were revealed to exhibit unique spectral signatures in this portion of the microwave spectrum. Generally, the brightness temperatures over the inland ice sheet were warmest at the lowest frequencies consistent with models that predict that those channels sensed the deeper, warmer parts of the ice sheet. Spectra along the lengths of outlet glaciers were modulated by the deposition and erosion of snow, driven by strong katabatic winds. Similar to previous experiments in Greenland, the brightness temperatures across the frequency band were low in crevasse areas. Variations in brightness temperature were consistent with spatial changes in sea ice type identified in satellite imagery and in situ ground penetrating radar data. The results contribute to a better understanding of the utility of microwave wide-band radiometry for cryospheric studies and also advance knowledge of the important physics underlying existing L-band radiometers operating in space.

## 1  Introduction

Because Earth's polar regions are vast, remote and characterized by extreme environmental conditions, satellite sensors have been used extensively for monitoring changes in the surface properties, volume, and extent of polar ice. Microwave sensors, both active and passive, are particularly suitable for this purpose due to their insensitivity to solar illumination and cloud cover.



Since microwave penetration into polar ice increases with decreasing frequency, low microwave frequencies are required to observe inner properties of sea ice and ice sheets. Until the early 2000s, the lowest frequency available on polar orbiting satellites was C-band (5.3 or 6.8 GHz for active or passive measurements, respectively) that limited ice sheet investigations to the properties of the upper 100 m of ice (Macelloni et al., 2007). With the launch of the L-band (1.4 GHz) SMOS (Kerr et al., 2010), Aquarius (Le Vine et al., 2010), and SMAP (Entekhabi et al., 2014) microwave radiometer missions, studies

of deeper properties became possible. It is estimated that the 1.4 GHz brightness temperatures observed are sensitive to ice sheet properties down to 900 m (Macelloni et al., 2016). Over first year sea ice (FYI), properties within 50 cm depth affect the L band emission (Johnson et al., 2021). Estimations of sea ice thickness – SIT (Kaleschke et al., 2012; Tian-Kunze et al., 2014) and of ice sheet internal temperature profiles (Macelloni et al., 2016, 2019) have both been demonstrated using L band brightness temperature measurements, although parameter estimations beyond the L-band penetration depth are extrapolations

subject to increased uncertainty. Consequently, brightness temperature observations at longer wavelengths are a motivation to probe even further into sea ice and ice sheets. That said, the use of longer wavelengths is also challenged by the fact that the corresponding lower frequency ranges are allocated to other services by international regulations, e.g. (US FCC allocation table, 2021), leading to radio frequency interference (RFI) for radiometric measurements.

Second, existing algorithms for geophysical parameters estimation are based on volume scattering which is negligible at

low microwave frequencies. Thus mew models are needed to interpretate brightness temperature measurements and retrieve geophysical parameters.

The development of advanced techniques for filtering RFI (as demonstrated for the 1400-1427 MHz band in the SMAP mission; (Mohammed et al., 2016)) has enabled the use of lower frequencies for monitoring polar regions even in the presence of other co-existing electromagnetic systems. A first airborne prototype (the Ultra-WideBand software defined RADiometer -

UWBRAD) that observes brightness temperature spectra in the range 500-2000 MHz (Andrews et al., 2018) was developed under a NASA Instrument Incubator Program led by The Ohio State University. Two airborne campaigns were successfully conducted in Greenland in 2016 and 2017 that demonstrated the feasibility of ice sheet temperature retrieval, the potential in distinguishing glacier facies, and the retrieval of information on sea-ice thickness and salinity (Andrews et al., 2018; Jezek et al., 2019, 2022; Yardim et al., 2022a). These campaigns also proved that polar geophysical measurements were possible in

this frequency range when advanced RFI processing is used to enable "opportunistic" brightness temperature measurements in RFI-free portions of time and frequency space (Duan et al., 2022).

The success of the Greenland campaigns motivated a new set of UWBRAD measurements in Antarctica. With support from the Italian Antarctic Programme (PNRA) and supplementary support from NASA, the "Ice Sheet and Sea Ice Ultrawideband Microwave Airborne eXperiment – ISSIUMAX" project was conducted in the austral spring of 2018. The project goals were (i)

to provide additional demonstrations of the use of 500-2000 MHz brightness temperature spectra for deriving vertical ice sheet temperature profiles in the inner part of Antarctica, and (ii) to demonstrate the capability of inferring glaciological information in coastal regions, including sea ice. An additional goal was to provide additional demonstration that microwave radiometry is feasible in the heavily used 500-2000 MHz spectral range in remote regions such as Antarctica. The ISSIUMAX campaign took place between October 30[th] and December 8[th], 2018, from the Italian Antarctic base (Mario Zucchelli Station) on the Ross Sea.





The campaign included the collection of airborne UWBRAD radiometer data over more than 3000 km of flight distance that was complemented by additional airborne and ground-based GPR (Ground Penetrating Radar) measurements and by satellite observations. Brogioni et al. (2022) reports and analyzes observations acquired in the inner part of the East Antarctic Plateau, and Andrews et al. (2021) describes in detail the instrument design and RFI processing.

This paper presents an overview of the campaign with a special emphasis on data collected in the coastal regions. The next
section provides a brief description of UWBRAD, and the sites surveyed are summarized in Section 3. The planning and execution of the campaign are detailed in Section 4, and Section 5 describes the 500-2000 MHz spectral variations observed for a variety of geophysical scenes of interest.

## 2   UWBRAD dataset description

UWBRAD is the first ultrawideband radiometer measuring brightness temperatures simultaneously over the 0.5-2 GHz range
through the use of twelve ~88 MHz bandwidth contiguous channels centered at frequencies from 560 to 1950 MHz and a circularly-polarized nadir pointed conical log spiral antenna (Andrews et al., 2018). The antenna is designed to have an approximate 60 deg half-power beamwidth that is independent of frequency, so that all frequency channels observe a similar region on Earth's surface. This beamwidth results in a 3 dB footprint on Earth's surface whose diameter is approximately 15.5% greater than the aircraft altitude above the local terrain. For the ISSIUMAX campaign, the nominal flight altitude was
in the 500-600 m range, and the average altitude was 541 m, resulting in a typical footprint diameter of 625 m. At the nominal flight velocity of 130 knots, this footprint remains Nyquist sampled (i.e. 2 measurements per footprint) at an integration time of approximately 4.5 s.

The UWBRAD radiometer performs two different kinds of calibration. The internal calibration process described in Andrews et al. (2018) is used to compensate for the impact of changes in system temperature or receiver gains on measured data by means
of the measurements of internal known loads. An external absolute calibration is then performed for each flight based on the use of known targets, i.e. the coastal moraine and the open ocean as hot and cold reference targets respectively. Before applying the external calibration, RFI detection is performed through a combination of algorithms as detailed in Andrews et al. (2021). A description of the UWBRAD calibration process and RFI detection is provided also in Appendix A.

The datasets shown in what follows are integrated over 5 samples representing an integration time of ~5 seconds. Studies
of the resulting 12 channel brightness temperatures over relatively homogeneous scenes show local variations having standard deviations from ~0.25 to ~1 K that vary with frequency and with the RFI environment at the time of the measurement. However, because the final external calibration of each frequency subchannel remains uniform over an entire flight duration, any instabilities in the instrument frequency response can impact the frequency response of the resulting calibrated brightness temperatures. Such effects are estimated to be within ±5 K based on an extensive analysis of the final dataset.
Figure 1 provides an example of the resulting 6144 point frequency spectra for 21:05 - 21:35 UTC on Nov 24[th] (i.e. 10:05 - 10:35 Terra Nova Bay time on Nov 25[th]) before (upper left) and after (upper right) the iterative RFI filtering and external calibration procedure. The correction or filtering of multiple spectral anomalies is evident, along with the associated loss of





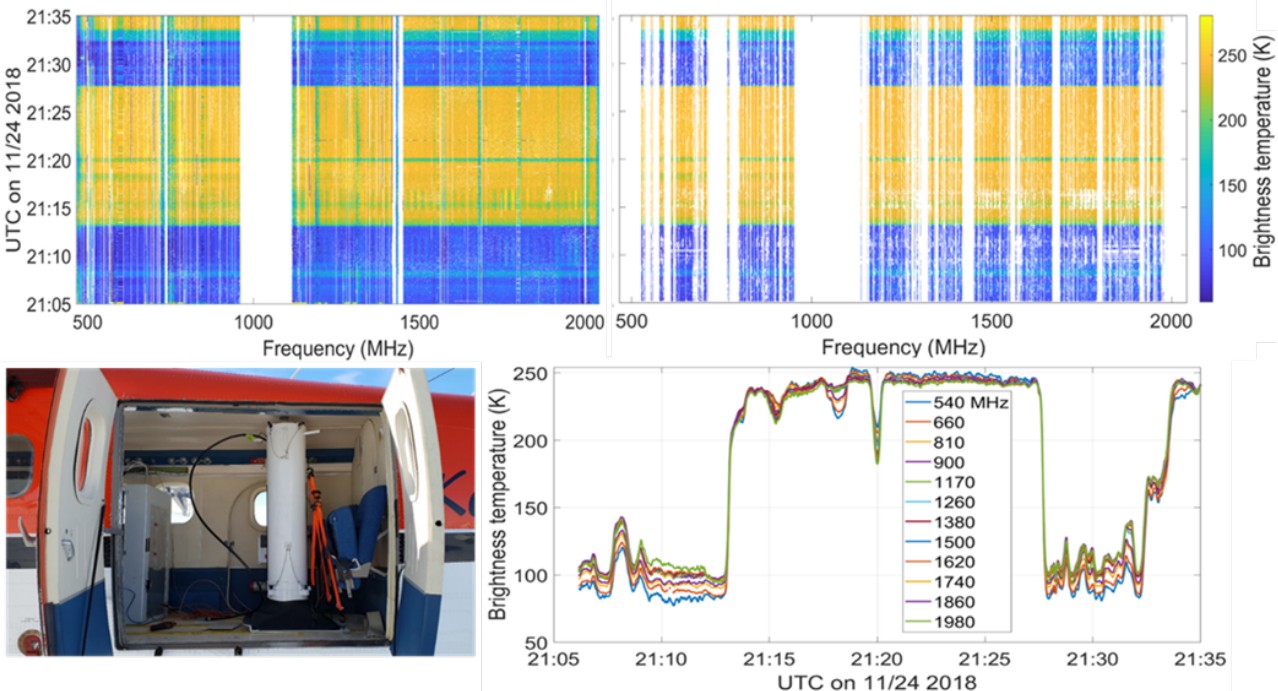

**Figure 1.** (upper) 6144 frequency channel spectrograms of brightness temperatures (Kelvin) before (left) and after (right) application of the iterative RFI filtering and external calibration procedure. (lower left) Photograph of UWBRAD antenna periscope and equipment rack aboard Twin Otter aircraft on Ken Borek Airlines. (lower right) Twelve channel integrated brightness temperatures corresponding to the spectrograms shown in the upper plots.

some portions of the measured spectrum for use in producing the final integrated products. The corresponding 12 channel integrated products for this time period are illustrated in the lower right plot and show transitions from open water to coastal

ice scenes having a variety of spectral features as described in what follows. The lower left portion of Figure 1 is a photograph of UWBRAD after installation on the Twin Otter aircraft operated by Ken Borek airlines. Visible in the photograph are the "periscope" system from which the conical spiral antenna is deployed after takeoff, as well as the electronics rack containing the UWBRAD radiometer receiver and data processing computers.

## 3 Test sites

The ISSIUMAX campaign took place in Victoria Land, East Antarctica where the Italian Antarctic Programme – PNRA operates two stations: Mario Zucchelli Station - MZS (74.695°S, 164.114°E) situated in Terra Nova Bay – TNB on the western coast of the Ross Sea, and Concordia Station (75.1°S, 123.33° E, operated in cooperation with the French Polar Institute Paul-Émile Victor – IPEV) located at Dome C on the East Antarctic plateau at an elevation of 3233m. Both stations were used as starting points for UWBRAD flights, but the analyses reported in this paper are for measurements collected in the coastal



region only. Preliminary results obtained on the ice sheet plateau further inland and close to Concordia station are reported in
(Brogioni et al., 2022).

Terra Nova Bay is bounded by the Drygalski Ice Tongue to the south, Cape Washington to the north, the Eastern Antarctic
plateau to the west, and the Ross Sea to the East. MZS is located on the coastal Northern Foothills range and is separated from
the ice sheet by the Transantarctic mountains (Figure 2). The geography contains many glaciological features of interest in the
campaign, including the Priestley, Reeves, and Campbell Glaciers (which feeds the Campbell Glacier Tongue), David Glacier
and the related Drygalski Ice Tongue, and sea ice around Cape Washington. The former two glaciers feed the Nansen Ice Sheet
which is actually an ice shelf, and is referred to as such throughout the paper for sake of clarity. The mean annual temperature
in the region is -14 °C, with January (mean temperature -2 °C) the warmest and May or August (-23 °C) the coldest months
(Frezzotti et al., 2001). Sea ice in the region is usually first year fast ice (FYI) typically 2 m thick close to the coast (Rack
et al., 2021) and that vanishes in December due to a combination of processes: late spring air temperature increases the ice
temperature and the brine volume that acts to weaken the fast ice (Frankenstein and Garner, 1967)(Weeks and Ackley, 1982),
then long-wavelength sea swell due to off-shore storms that break the pack which in turn is moved offshore by the katabatic
winds, e.g. (Bromwich and Kurtz, 1984).

Occasionally, fast ice can remain in place for more than one year in some sites, for instance in the inner parts of Silverfish
Bay, Wood Bay or Tethys Bay. In this case the multi-year land fast ice (MYI) can reach higher thickness of 3-4 m as encountered
in the campaign.

## 4  Campaign planning and execution

Flight routes were planned based on the flight time assigned to the project (35 flight hours divided among radiometric and
ground penetrating radar measurements) and coverage of:

- sea ice in Terra Nova Bay and Wood Bay,

- the Nansen Ice Shelf and related inlet glaciers,

- the ice sheet along paths from MZS to Concordia Station and MZS to Talos Dome (73.0°S, 158.0°E)

- other secondary targets such as buried lakes, open sea, exposed outcrops (e.g., nunataks and moraines), and blue ice.

Sea ice flights were planned based on the known presence of grease ice/nilas bands and fast ice as well as the expectation of
highly stable FYI and minimal multi-year ice (about 1.5-2 years) at these locations. MYI was identified by analyzing long
timeseries of MODIS and Sentinel-1 images. Complementary flights of a helicopter-mounted 400 MHz Ground Penetrating
Radar (GPR) were made in order to achieve additional information about the electromagnetic response of the sea ice and its
snow cover.

The final plan specified six flights, of 5 h each. Flight 1 was planned to focus on sea ice and included many transects over
the Gerlache Inlet, Silverfish Bay and Wood Bay (Figure 2). The transects were designed to survey sea ice in directions both

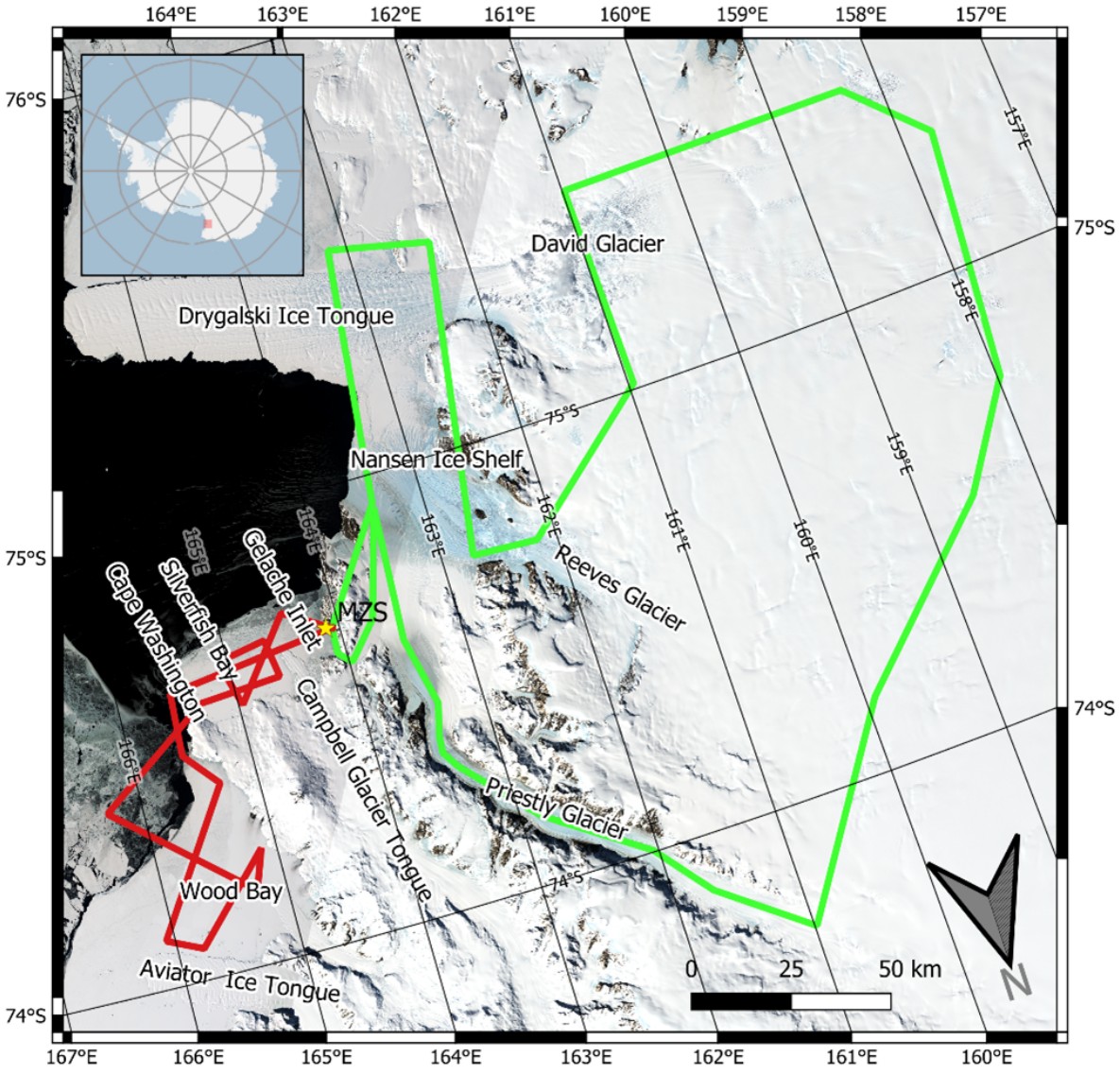

**Figure 2.** Map of Terra Nova Bay. The planned routes of flights 1 and 2 are represented in red and green respectively. Yellow star indicates the location of Mario Zucchelli Station (MZS). The background image is a mosaic of Landsat 8 RGB composites collected on November 24[th] (coastal zone, contemporary to the flight) and November 27[th] (inland portions).

parallel and orthogonal to the coast in order to capture the variability of the sea ice thickness (SIT) and ice type. The inner part of Wood Bay is characterized by very thick fast ice (FYI or MYI of approximately 3 m thickness), whereas the outer part of





the bay contains FYI of ~2 m thickness. The external part of the bay in contrast hosts nilas and grease ice. Sea ice in Wood Bay is usually snow-covered. Silverfish Bay and the Gerlache Inlet instead are covered almost entirely by 2.0-2.5 m thick FYI,

and usually experience different dynamics given the katabatic winds at these locations that remove snow from the surface.

Flight 2 was planned primarily to observe terrestrial ice inland from MZS. The flight surveyed the Priestley Glacier along the route of a 2013 Operation Ice Bridge (OIB) survey (from which information on the inner structure of the glacier can be obtained), and then proceeded over the ice sheet toward the David Glacier. Results from Flights 1 and 2 are the subject of this paper.

Flight 3 was designed to collect data over the Antarctic ice sheet and followed the route of the ITASE 98/99 traverse. Flight 4 was planned to collect data over glaciological features in the vicinity of Concordia station, including Aurora Basin (150 km north of Concordia Station) which is characterized by deep trances in the bedrock and a network of subglacial lakes such as Concordia Lake. Little Dome C, the site of the Beyond EPICA – Oldest Ice drilling, was also surveyed (Augustin et al., 2004) for which, topography is known in detail (Lilien et al., 2021), and several temperature profiles of the ice are available (Ritz et

al., 2010). The return Flight 5 from Concordia Station to MZS was planned to survey the megadune region in the East Antarctic plateau (Frezzotti et al., 2002; Courville et al., 2007) , multiple blue ice sites, and buried lakes in the coastal region. Brogioni et al. (2022) provides a detailed description of these inland flights.

The campaign was organized in phases. The first phase included the collection of GPR surveys and in situ measurements of sea ice (Nov 3rd – 18th). Acquisition of radiometric data for the coastal area (Nov 24th - 25th) followed, along with sky

calibration measurements (Nov 29th - 30th). Finally, measurements were completed over the inland ice sheet (December 2nd - 5th).

## 4.1   GPR surveys

The activities scheduled for the first part of the campaign were carried out using a GPR instrumentation (GSSI Sir3000) equipped by a 400 MHz helicopter mounted antenna. The instrument has a 150 ns investigation range, with a 16 bit sampling.

The helicopter averaged speed was about 40 knots which implies the acquisition of a single trace about every 28 cm while calculated vertical resolution (into the ice) was about 32 cm. Considering that flight height was about 4-5 m, the time range allowed an ice penetration of about 10-11 m. The aim of GPR survey was to collect measurements of sea ice thickness in the Gerlache inlet, Silverfish Bay, Wood Bay, and the ice structure along a path between the Nansen ice shelf and the Priestley Glacier (Figure 4). The planned GPR flight lines were to be repeated at least three times to study any progressive increase

in brine volume of the ice as it warmed (Frankenstein and Garner 1967). Preparation and assembly of the GPR and related instruments occurred from Oct 30th to Nov 2nd, and on Nov 3rd the first survey covering the area between MZS and Cape Washington was acquired (Leg1: TNB and Silverfish Bay; shown in red in Figure 3). Cloud cover continuously present in the region until Nov 7th that prevented flights until Nov 9th when measurements were performed over Wood Bay (Leg2, purple line in Figure 3) and repeated for Leg1. A flight over Leg3 (blue line in Figure 3) was performed on Nov 12th, followed by a repeat

of Leg1, Leg2 and part on the sea ice of Leg3 on Nov 18th.



The survey benefitted of the presence of several sea ice coring holes in Silverfish Bay and Gerlache Inlet that were used as reference for depth conversion of GPR data. In addition, on Nov 18th sea ice and sea water were sampled in Silverfish Bay for salinity measurements by using a COND61 XS Instruments Italy (calibrated with proper standard solutions). The results showed sea ice salinities of 6.14, 6.35 and 7.56 g kg$^{-1}$ at 0.6, 2, and 2.8 m depths respectively, while the sea water below the

ice ranged around a value of about 31.8 g kg$^{-1}$ (yellow triangle in Figure 3). Other sea water salinity measurements conducted in Gerlache Inlet showed values among 31.2 and 32.1 g kg$^{-1}$ (cyan triangles in Figure 3).

## 4.2    UWBRAD surveys

UWBRAD was installed on board the Twin Otter aircraft on November 24th, and the first and second flights were performed on November 25th. The flight's route included 41 waypoints plus a further 4 that overlapped ICESat-2 measurement tracks.

The aircraft departed MZS at 09:30 AM local time towards a polynya that had formed north of the Drygalski Ice Tongue (to enable open water calibration measurements), and measurements concluded at approximately 1:00 PM local time when the aircraft landed at MZS for refueling. During this flight, the air temperature recorded by the AWSs at MZS ranged from -4°C to -2°C. Other AWS sites, e.g. at Cape King (73.586°S, 166.621°E) and Boulder Clay (74.751°S, 164.021°E), reported 1°C - 2°C cooler temperatures. Flight 2 then began at 3:00 PM with the aircraft proceeding inland along the Priestly glacier. The

flight proceeded according to the planned route until reaching the Reeves Glacier when, due to adverse wind conditions in the region, it was decided to eliminate the two transects from the Nansen Ice Shelf toward the Drygalski Ice Tongue. Due to this change, it was possible to perform two other transects of sea ice in Wood Bay along the ICESat 2 satellite tracks. Flight 2 was completed with the return to MZS at 7:20 PM. The final flight paths achieved are shown in Figure 4.

On November 29th, the microwave radiometer was installed on the ground near the Oasi shelter (away from the main MZS

buildings) in order to perform sky viewing calibrations (Andrews et al., 2021). The data collected were used to perform calibration tests and studies of RFI. These measurements continued on November 30th with varying levels of additional attenuation placed on the radiometer antenna and concluded on December 1st.

Flight 3 towards Dome C began at 22:26 UTC on December 2nd and a transect over one of the ESA Domecair experiment routes was performed before landing at Concordia Station to allow intercomparisons with the 1.4 GHz brightness temperatures

previously acquired by the EMIRAD sensor (Kristensen et al., 2013). Flight 4 over Little Dome C and two other Domecair flight lines then occurred on Dec 3rd from 5:25 UTC to 9:32 UTC. The Flight 5 return to MZS was performed on December 5th. On this flight, UWBRAD acquired data only for the Mid Point - Terra Nova Bay route.

## 4.3    Summary of datasets

The final UWBRAD dataset of geolocated nadiral brightness temperatures in 12 frequency channels covers a distance of more

than 5200 km. The GPR dataset covers the sea ice region and was collected prior to the radiometric flights. However, no major sea ice dynamic events were observed prior to the coastal UWBRAD flights so that the two datasets can be reasonably compared. GPR data were acquired over sea ice having thickness 2 m or more that is expected to remain fast until major break-up events at the beginning of December.



**Figure 3.** GPR flight routes: Leg 1 in Terra Nova Bay and Silverfish Bay (red lines), Leg 2 in Wood Bay (purple lines), and Leg 3 from Priestley Glacier to the Gerlache Inlet (blue lines). Cyan triangles indicates the sea water salinity test sites, yellow triangle marks the site of the sea ice salinity measurements.

Datasets from other spaceborne remote sensing instruments were also assembled for intercomparison, see the detailed list

of data sources in Appendix B. Both optical (e.g. Landsat-8 and Sentinel-2) and radar (e.g. ALOS, Sentinel-1 and COSMO-

**Figure 4.** Map of Flight 1 and 2 as executed over the coastal regions. Colors represent the UWBRAD brightness temperature (Tb) measured at 560 MHz. The background image is a mosaic of Landsat 8 RGB composites collected on November 24[th] (coastal zone, contemporary to the flight) and November 27[th].

SkyMed) datasets in particular were collected over all the flight routes. Mosaics available through the Quantarctica collection (Matsuoka et al., 2021) were also examined, including the RAMP project Radarsat mosaic (Jezek, 1999; Jezek et al., 2013) and the Landsat Mosaic of Antarctica (Bindschadler et al., 2008). Ice sheet thickness information was also obtained from the Bedmachine project (Morlighem et al., 2020), and geothermal heat flux estimates from Fox Maule et al. (2005).





## 5 Results

The brightness temperature spectra acquired by UWBRAD respond to the properties of the medium observed. Expectations for properties of brightness temperature spectra and their trends versus frequency for common targets will be used to interpret the results that follow. For example, typical "water" and "thin sea ice" spectra show an increasing trend versus frequency given the dielectric properties of sea water and the decreased electromagnetic loss through thin sea ice at lower frequencies. "Ice sheet" spectra in contrast decrease with frequency due to the sensitivity of lower frequencies to the warmer ice at greater depths. Both cases can further be impacted by any scattering effects within the medium observed, which typically would be expected to decrease observed brightness temperatures and to have a greater impact on higher frequencies. The scatterers producing such effects must be of a size that is appreciable compared to the wavelength, i.e. of at least ~cm scales so that the impact of features such as snow grains can typically be neglected. Scatterers of these sizes are more likely to occur in portions of the snow/firn that experience periodic melt and refreeze events, as discussed in Jezek et al. (2017). All of these physical effects are evident in the initial examples presented in what follows. These results provide evidence of the information contained in measurements of brightness temperature spectra in coastal Antarctica, including scenes containing sea ice, glaciers, rocks, ice shelves, and subsurface lakes.

### 5.1 Sea ice

A first example highlighting the spectral behavior of different sea ice types comes from a Flight 1 transect that passed from Gerlache Inlet to Silverfish Bay (left plot of Figure 5). An analysis of a one-year timeseries of Sentinel-1 Synthetic Aperture Radar - SAR images showed that the near-coastal sea ice (or "fast ice") at sites 1, 3, and 5 in Figure 5 formed in March-April 2018. GPR measurements reported an ice thickness of 2.0-2.5 m. Katabatic winds near site 1 cause ablation/sublimation of the fast ice and result in almost no snow coverage, while the fast ice at sites 3 and 5 was covered by snow of less than 45 cm thickness (estimated from GPR data). The same Sentinel 1 analysis showed that the "outer region" sea ice at site 2 in contrast was the product of multiple breakup processes. This ice at the time of ISSIUMAX formed in September composed by old small floes and newly formed ice. The GPR survey showed an average thickness of only 1.5 m in this region along with evidence of relative increase in brine volume of the ice at site 2. The site 4 ice of the Campbell Glacier Tongue which is inland glacier ice which flows through the Transantarctic Mountains and then over the sea, therefore has a much greater thickness (about 400 m at the hinge point; Han and Lee, 2014) and distinctly different dielectric properties. Indeed, this continental ice is generated by the accumulation of snow on the plateau over centuries and can be considered to be pure ice with very small losses, as opposed to the sea ice of the other sites that is characterized by high dielectric losses due to the inclusion of saline brine pockets. These differences significantly impact microwave penetration through the ice and therefore the ice properties sensed by UWBRAD (Demir et al., 2022).

A time series of UWBRAD brightness temperatures is provided in the left portion of Figure 6 along with brightness temperature spectra at the five labeled sites in the right plot. As expected from theory (Demir et al., 2022; Johnson et al., 2021), the thicker sea ice at sites 1, 3, and 5 shows a "flatter" spectrum having brightness temperatures in the 240-250 K range. This is due





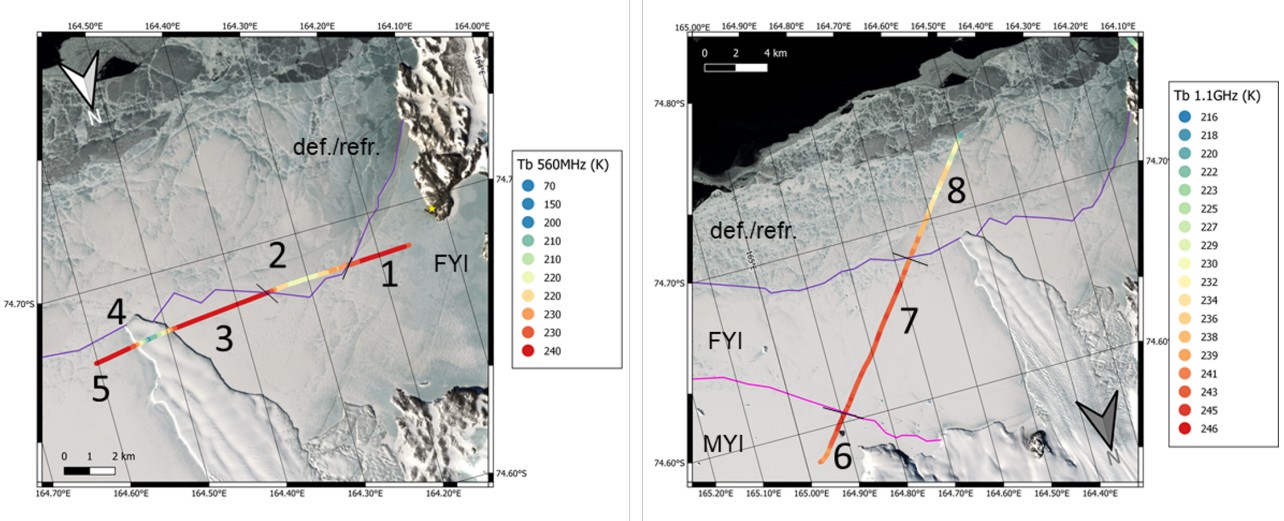

**Figure 5.** (Left) Map of the transit from Gherlache Inlet to Silverfish Bay. Fast Ice is present at sites 1 (FYI), 3 (FYI) and 5 (FYI), thinner deformed/refrozen ice at site 2, and site 4 marks the Campbell Glacier Tongue. (right) Map of transit from Silverfish Bay toward open sea. Pink and purple lines delimit MYI fast ice, FYI fast ice and deformed/refrozen sea ice. The base map of both panels is a Landsat 8 RGB composite acquired contemporaneously with the flight.

to the sensitivity of brightness temperature to ice thickness: for a salinity of 6 g kg$^{-1}$ (as measured in-situ) the electromagnetic - e.m. signal saturates for thicknesses higher than 1.5 – 2 m even at 500 MHz. The flattest spectrum occurs for the bare ice at

site 1, while the spectra of sites 3 and 5 show a slight decreasing trend with frequency. These small differences may be due to the impact of a depth hoar layer within the snow and/or differing temperature profiles inside of the ice caused by snow thermal insulation. However, routine measurements made by MZS on the sea ice near the airstrip revealed an ice temperature in the -5°C to -2°C range at depths to 30 cm below the surface, so that the ice was almost isothermal at least to 30 cm depth (MZS operations room, personal communication, Jan 15$^{th}$ 2022).

Over the more dynamic and thinner ice at site 2, brightness temperatures show the expected lower values associated with an increase with frequency that was also observed in UWBRAD's Greenland campaign where the ice was typically thinner than 2 m (Jezek et al., 2019). At site 2, the presence of higher brine volume within the ice also may impact this result because the greater penetration at lower frequencies results in a greater sensitivity to the higher brine volume seen at greater depths than at higher frequencies.

Brightness temperatures over the glacier tongue (site 4) decrease uniformly with frequency from ~210 K (560 MHz) to 182 K (1950 MHz). This behavior is typical of continental ice (Jezek et al., 2015; Yardim et al., 2022a; Brogioni et al., 2022) as described previously. The smooth transition of brightness temperatures observed in Figure 7 during the overpass of the ice tongue is due to the large antenna footprint of UWBRAD. The spectra of Figure 6 (right) were obtained by averaging acquisitions in the vertically shaded portions of Figure 6 (left) that correspond to approximately uniform regions in space.





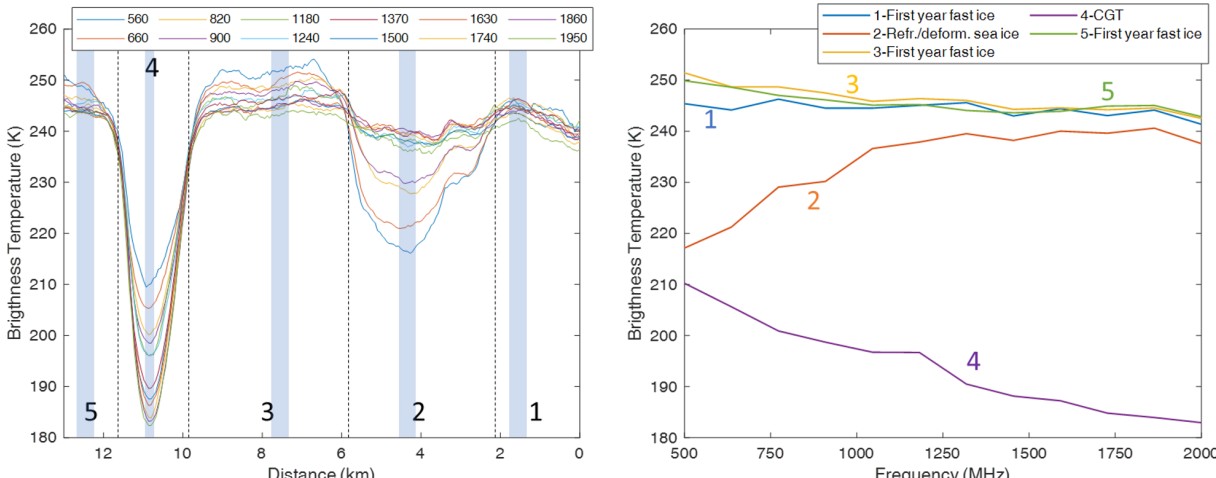

**Figure 6.** (left) Transect of UWBRAD brightness temperatures corresponding to the left portion of Figure 6; the legend indicates the frequency in MHz of the twelve UWBRAD channels. (right) Spectra of the five different targets obtained as an average of data shaded in blue.

The right portion of Figure 5 displays the flight path of a similar example that proceeds from Silverfish Bay toward the open sea beyond the tip of the Campbell Glacier Tongue (CGT). The transect begins over multi-year ice and proceeds toward the thinner ice to a refrozen sheet of pack/fast ice. The ice thickness along this path is estimated by GPR measurements to be in the range 2.0-2.5 m for the majority of the transect (Figure 7 bottom panel). As shown in Figure 7, brightness temperatures over the multi year (sites 6, 0-4 km along the flight path) and first-year fast ice (site 7, 4-14 km) are near 245 K, although the MYI tends

to be slightly cooler than the FYI fast ice (note that some channels in the earlier part of the plot are missing due to a reboot at this time of one of UWBRAD's data acquisition computers). For both site 6 and 7, the ice thickness estimated by GPR is greater than 2 m, so that brightness temperatures are again saturated as for similar sites in Figure 6. As in the previous example, lower frequency channels show slightly higher brightness temperatures (by ~2-4 K) than higher frequencies. GPR measurements over site 6 (shown in the middle portion of Figure 7 and interpreted into snow and ice thicknesses in the lower panel) confirm

the presence of multi-year ice (nearly [nd] year ice, dated by using satellite images) for which the scattering within the ice or at the ice/water interface was quite high, so that brightness temperatures are moderately lower than those at site 7 where GPR measurements show much lower evidence of scattering. Differences between these two sites may also be related to a lower brine content in the MYI at site 6 as compared to the FYI at site 7. This is confirmed also by measurements collected over Wood Bay (Figure 8) where the Tb of MYI (identified by pink contours) is cooler than Tb over first year fast ice (delimited

by purple line). Brightness temperatures at site 8 are similar to those at site 2 and there is evidence of increased scattering and the percolation of sea water into the snow-ice interface. As at site 2, brightness temperatures increase with frequency likely due to the infiltrated water's impact. SAR backscatter measurements from Sentinel-1 (C-band) and COSMO-SkyMed (X-band) are also included for this transect, and show a response that is negatively correlated to brightness temperatures as



**Figure 7.** (Top) Timeseries of UWBRAD and SAR data as a function of position along the path in the right plot of Figure 6. (Middle) echogram obtained by GPR along with visual interpretation. (Bottom) Sea and snow thickness estimated from GPR data. Some UWBRAD data are missing in the first 8km due to an unexpected computer reboot.

should be expected when either surface or volume scattering effects are present within the ice observed (although the influence

of particular scatterers should be expected to increase with frequency).
**Figure 8.** Map of the transects over the inner part of Wood Bay. Pink and purple lines delimit MYI (2 years) and FYI fast ice respectively. The base map is a mosaic of COSMO-SkyMed X-band SAR images acquired on Nov 21st and Nov 27th 2018 (4 days before and 3 days after the flight respectively).

A third example was acquired in Wood Bay as the aircraft moved from open water to fast ice passing through a grey/white ice field (Figure 9, left). As discussed previously, sea ice in Terra Nova Bay in November-early December is characterized by near-coastal FYI fast ice of 2-3 m thickness surrounded by nilas and gray ice in some cases (Mezgec et al., 2017; Brett et al., 2020). Occasionally MYI is present very close to the coast. The path shown begins over open water (low brightness temperature), then enters the grey/white ice region, and finally arrives over thick ice of about 2.5 m thickness. Over this transect the ice was free of snow. Brightness temperature time series along this path are shown in the upper left plot of Figure 10, with spectra for



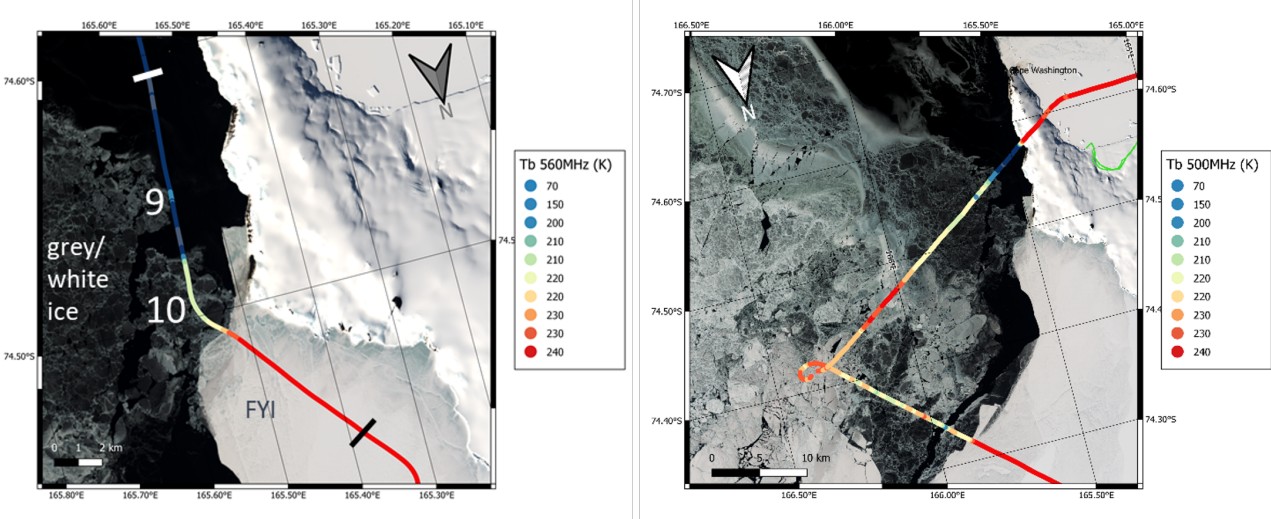

**Figure 9.** (left) Map of the transit from Cape Washington to Wood Bay, white and black marks indicate the beginning and end of the transect, respectively. (right) Map of the transit from Wood Bay to Cape Washington. The base map is a Landsat 8 RGB composite acquired contemporary to the flight.

selected locations shown in the upper right plot. The difference between site 9 and site 10 appears to be influenced by the ice concentration within the UWBRAD footprint and potentially any differences in ice thickness at these locations. Spectra for the two sites show an increasing trend similar to that observed for open water, but having a mean value that increases as the ice

concentration increases and a saturation at a frequency that shifts toward lower frequencies as the ice concentration increases. The spectrum for the thicker ice in this region is flat with an average brightness temperature of 240 K as for the snow-free ice at site 1 in Figure 5.

A final sea ice example includes a passage from fast ice of 2.5-2.7 m thickness to a young ice field as the airplane returned to Silverfish Bay from Wood Bay (Figure 9, right). The background image in this plot is a Landsat 8 RGB composite acquired

two hours before the Wood Bay survey. Given the weak sea currents in the bay expected at this time, the image provides a reasonable representation of the ice at the UWBRAD acquisition time. The brightness temperature timeseries for this transect is shown in the lower left plot of Figure 10, along with the panchromatic and thermal IR data from the Landsat 8 product. The high correlation observed between the panchromatic data and brightness temperature variations provides further evidence that the Landsat 8 image is suitable to support the interpretation of UWBRAD acquisitions. Depending on the ice concentration

and thickness, brightness temperatures vary from ~120 to ~240 K. Note that young ice is typically much more saline than FYI, reaching salinities up to 12-15 g kg$^{-1}$ (Cox and Weeks, 1974) so that brightness temperatures saturate for shallower ice thickness. By using Landsat 8 IR data and the thermodynamic model described in (Maykut and Untersteiner, 1971; Yu and Rothrock, 1996; Drucker et al., 2003) it is possible to estimate ice thickness in the young ice field. The air temperature, required for this analysis, was derived from the MZS AWS and the ECMWF ERA5 reanalysis (Hersbach et al., 2018, 2020),

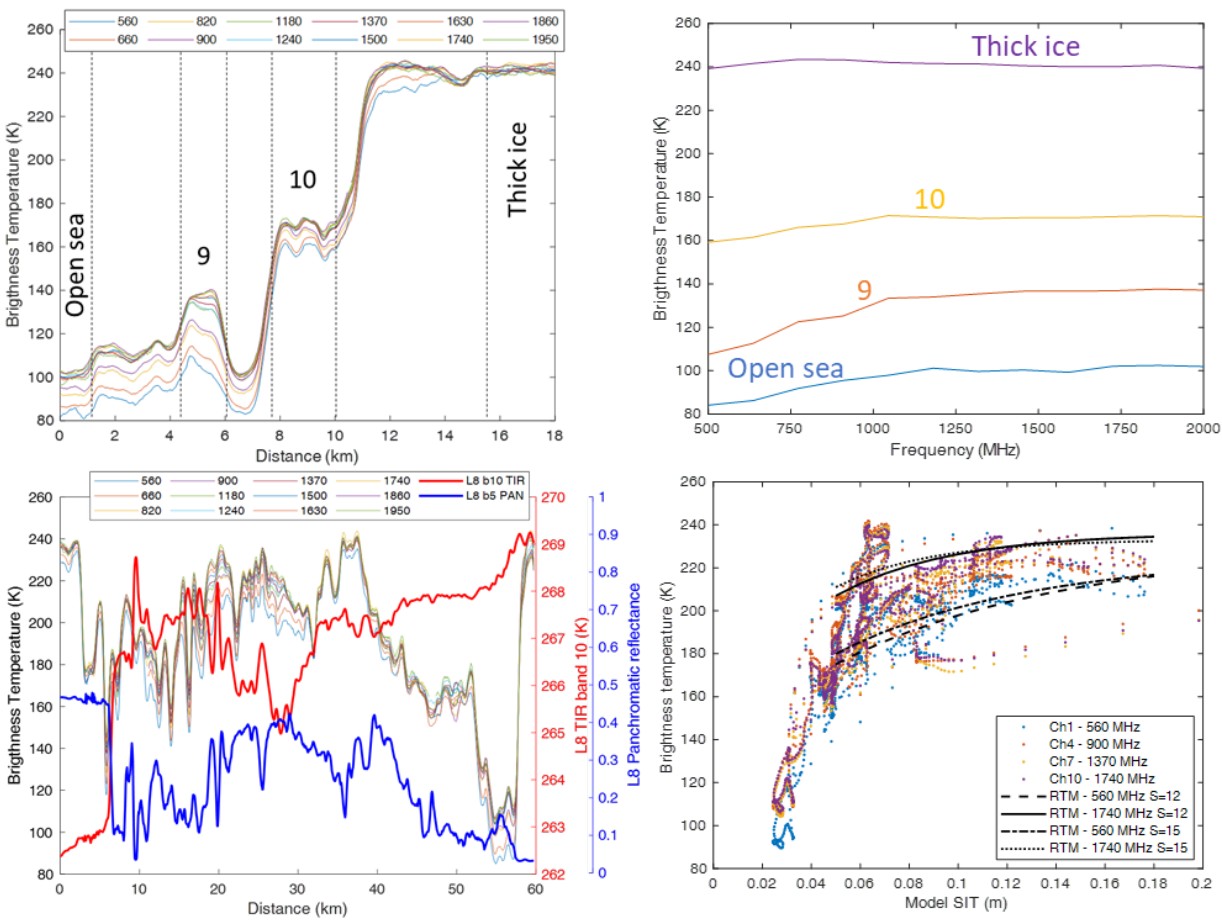

**Figure 10.** (upper left) Timeseries of brightness temperature corresponding to the left plot of Figure 9. (upper right) Spectra of four selected targets from the left plot of Figure 9. (lower left) Timeseries of UWBRAD brightness temperature, surface temperature derived from band 10 TIR of Landsat 8 (red), and reflectance derived from band 8 panchromatic of Landsat 8 (blue) for the transect in the right plot Figure 9. (lower right) Scatterplot of UWBRAD channels 1,4,7,10 versus estimated sea ice thickness along with e.m. radiative transfer model estimation for $T_{ice}$ = -4°C, $T_{water}$ = -2°C and salinity = 12 g kg$^{-1}$ (continuous and dashed lines) and 15 g kg$^{-1}$ (dotted and dash-dotted lines).

while the superficial ice temperature was obtained from the Landsat TIR bands. Note that this method is applicable for ice up to approximately 50 cm thickness in stationary conditions. Moreover, since a validation of the retrieved ice thickness is not possible, we consider that values obtained as a proxy for assessing the dependence of UWBRAD brightness temperatures on SIT.

Figure 10 (lower right) plots the UWBRAD brightness temperatures in four frequency channels versus the retrieved ice thickness; the predictions of a radiative transfer model for brightness temperatures are also included. As expected a rapid increase in brightness temperature at all frequencies versus ice thickness is observed. It should be noted that shallow young



ice has a brightness temperature similar to thick FYI due to its increased brine content and electromagnetic losses. Young ice is shallower than FYI but with a higher salinity, resulting in similar brightness temperatures. While the four frequency channels shown appear to respond similarly, at 560 MHz, a lower brightness temperature is reached at the maximum thickness

as compared to the other higher channels, suggesting that further sensitivity to ice thickness is available (though not for the limited ice thickness available along this transect). The curves in Figure 10 (bottom right) represent simulated data that were obtained by using an electromagnetic radiative transfer model (Picard et al., 2013): the ice is assumed to be a homogeneous layer whose permittivity is obtained from Vant et al., (1978) overlying a semi-infinite medium representing sea water (Klein and Swift, 1977). The ice temperature is assumed -4°C (average temperature recorded by the AWS at MZS and Cape King)

while the ice salinity is 12 and 15 g kg$^{-1}$ (typical values for young ice). Simulated trends show reasonable agreement with the sensitivity of the experimental data (better for a salinity of 12 instead of 15 g kg$^{-1}$) indicating that the sensitivity of brightness temperature to ice thickness is strongly affected by the brine content of the ice (with higher ice salinities reducing sensitivity to ice thickness as expected). This result has implications for SIT estimation from satellite measurements, and confirms the importance of ice salinity information in the thickness retrieval process.

All the examples previously described illustrate the sensitivity of 500-2000 MHz brightness temperature spectra to a variety of sea ice properties including ice thickness, salinity, snow cover, and the presence of scattering or higher brine volume (driven by higher salinity) within thin ice. Because of the natural heterogeneity of sea-ice these effects should be expected to be more significant in the ~625 m footprint of UWBRAD as compared to the 40-60 km spatial footprints obtained from the current satellite observations of missions such as SMOS and SMAP.

**5.2 Glaciers**

In the Terra Nova Bay region, glaciers flow outward from the Transantarctic mountains toward the sea. For instance, Priestley Glacier flows for about 120 km from the eastern part of Victoria Land into the Ross Sea and feeds the Nansen Ice Shelf. Priestley Glacier was surveyed in 2013 by OIB's Multichannel Coherent Radar Depth Sounder (MCoRDS) and Accumulation Radar. The UWBRAD route of Flight 2 was aligned with these OIB transects (Figure 11). Given Priestley Glacier's velocity

of about 100 m/yr (Frezzotti et al., 2000; Mouginot et al., 2012), a displacement of about 500 m should be expected between the OIB and UWBRAD data. This displacement, which corresponds roughly to one UWBRAD footprint, is neglected in what follows.

Figure 12 compares UWBRAD brightness temperatures with the MCoRDS radargram along this path. Significant decreases in brightness temperatures are found to occur over highly crevassed areas identified in optical imagery from the aircraft (sites

a to e) that also show high backscattering values for MCoRDS, Paden et al. (2010), (as well as the OIB Accumulation Radar, not shown here, Paden et al. (2014)) and spaceborne SAR data (ALOS L-band data shown in Figure 12). In these regions, UWBRAD brightness temperatures show only a weak dependence on frequency (smaller than 1 K/GHz). This is expected since the apparent roughness of these zones is large for all the wavelengths considered. The remaining regions along the glacier can be divided into two subtypes. Brightness temperature spectra in the upstream section (site 12, at distance 115-

125 km along the transect) decrease with frequency by about -5 K/GHz as typically observed for ice sheets. In contrast, blue

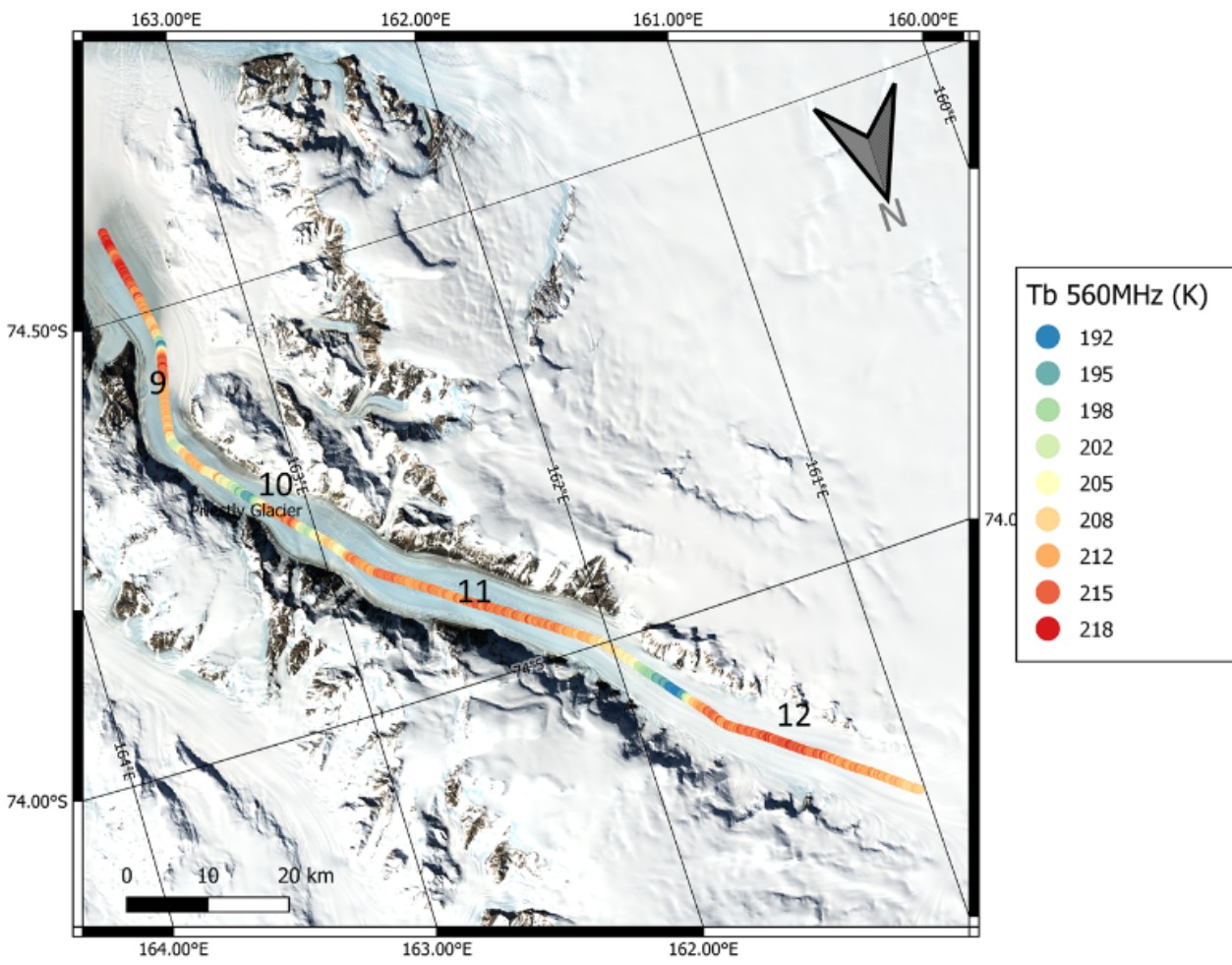

**Figure 11.** Map of the transit over the Priestley Glacier along the 2013 OIB route. Numbers indicates sites highlighted in Figure 12. Base map is a Landsat 8 RGB composite.

ice portions of the glacier show an opposite trend up to a maximum slope of +7 K/GHz. This is particularly evident in the measurements at distance ranges 25-30 km, 45-50 km and 55-65 km (sites 9, 10, 11 respectively). This behavior may be due to different temperature profiles within the glacier in the two different regions. Upstream the structure of the ice is similar to that further inland, with about 80-100 m of snow/firn above the ice layers. Inside the canyon, katabatic winds ablate the snow/firn 340 layers, leaving exposed ice having a lower albedo and higher thermal transmissivity (bluish color of the glacier in Figure 11). The transport of ice along this rapidly moving glacier also can influence the temperature profile within the ice.





**Figure 12.** Comparison between UWBRAD data collected over Priestley Glacier (upper) and MCoRDS acquisitions (lower, colorscale indicates the intensity of the MCoRDS radargram) . The continuous blue line indicates the surface of the glacier while the glacier bedrock is marked in red. The vertical red dashed line marks the glacier grounding line.

Ice shelf brightness temperature spectra (at distances 0-25 km in Figure 12) contrast with ice upstream of the grounding line (sites 9 and 10) marked by the red dashed line in Figure 12. The spectrum over the ice shelf shows a spectral variation of 25 K with frequency as compared to only 14 K over the glacier.

**Figure 13.** Map of transects over the Nansen Ice Shelf. The colorscale indicates the 560 MHz brightness temperature, while the background image is C-band SAR data acquired by Sentinel-1 in November 2018. The green curves indicate the grounding line as derived from MEA-SURES dataset, while sites 13 and 14 are snow accumulation areas described in the text.

## 345  5.3  Nansen Ice Shelf

UWBRAD measurements were also performed over the Nansen Ice Shelf (which is fed by the Priestley and Reeves Glaciers). Due to katabatic winds that hampered the flight, only the northern portion was surveyed. The 560 MHz brightness temperatures shown in Figure 13 point out a variety of features in this region. For example, in the final section of the Reeves Glacier, just upstream from the grounding line, 560 MHz brightness temperatures show strong variations, with minima around 170 K over





crevassed areas and maxima around 230 K over more compacted ice regions. These values are in line with those measured over the Priestley Glacier (Figure 12). The 560 MHz brightness temperatures over the ice shelf in contrast are quite high, in the range 220-240 K. Additional variations are noted on the path from Reeves Glacier to Inexpressible Island that appear highly correlated to the C-band backscattering variation of the Sentinel 1 background image.

UWBRAD brightness temperatures for site 13 are compared in Figure 14 with L-, C- and X-band SAR measurements as
well as the GPR-measured radargrams. The Sentinel-1 SAR image indicates the presence of a snow accumulation region at site 13 at the end of the Priestley Canyon (ALOS and COSMO-SkyMed images have a similar pattern, not shown here). Before and after the snow zone, brightness temperatures are quite high, on the order of 225 K for all channels, and drop to values ranging from ~185 K (560 MHz) to ~145 K (1950 MHz) in the snow-covered area. This behavior is inversely correlated with the SAR backscattering variations, which show higher backscatter over the snow and lower over the blue ice region. At UWBRAD
frequencies, dry snow is expected to be transparent, with the dominant effect being the weak multiple reflections between layers of different density, e.g. (Brogioni et al., 2015). The snow zone encountered here appears to represent a high-scattering layer of maximum 20 m thickness (estimated from MCoRDS, in GPR data the depth of scattering area exceeds the investigation range which is about 12 m), buried below a 1 m layer of non-scattering snow. The most reasonable explanation for this behavior is the seasonal melt/refreeze cycles of the snow in which liquid water percolates into the snowpack to form larger ice lenses and/or
columnar ice inclusions that cause increased scattering. Given that the snow area is not eroded by the winds, the inclusions have grown sufficiently large to cause scattering at these low frequencies.

This behavior is similar to that observed by Jezek et al. (2017) in the percolation zone inland of Nunatarsuaq, Greenland. Note that lower frequencies are expected to be less affected by scattering than higher frequencies, as observed in the measured brightness temperature time series in Figure 14. Air temperature measurements collected by the AWS Sofia (74.8167°S,
163.2333°E, very close to the snow area) over the period 1987-2002 show that air temperatures rose above 0°C for several days almost every year. A similar analysis of data from AWS Zoraida (74.16°S, 162.7392°E, located 40km upstream of site 13 at an elevation of 880m msl - about 800m more), showed that the air temperature rose above 0°C at least 3 times in the last 30 years. An analysis of the melt presence indicator (Torinesi et al., 2003) applied to the K-band brightness temperatures from Advanced Microwave Scanning Radiometer 2 (AMSR2, Japan Aerospace Exploration Agency, 2013) confirms that seasonal
melt can occur in some years in this region, so that the presence of larger scatterers is plausible.

A similar behavior is noted also at the confluence between the O'Kane and Priestley Glaciers at site 14 (Figure 13), where GPR measurements again show a higher scattering layer beneath an upper snow accumulation layer. This effect could again be explained by the melt/refreeze of snow accumulated by turbulence of the katabatic winds in the final bend of Priestley canyon.

## 5.4 Rocks

Exposed rocks are another interesting target for UWBRAD since they represent one of the "hottest" brightness temperature objects observed. In Terra Nova Bay, "rocks" consist mainly of granite and volcanic layers that appear dark in optical imagery. During Flight 2, two sites were surveyed that were sufficiently large to cover a UWBRAD footprint: a nunatak in the David glacier region (75.47340°S, 159.60°E, 1315 m a.s.l.) and a portion of Inexpressible Island (74.8780°S, 163.6408°E, 258 m







**Figure 14.** Comparison between UWBRAD and SAR data collected across the snow zone on the Nansen Ice Shelf (top) and GPR acquisitions (bottom). Red colorscale indicates the intensity of GPR radargram. Shaded area indicates data collected over site 13.

a.s.l.). Surface temperatures derived from Landsat 8 for these locations were -15°C and -9°C respectively, which is consistent
with a moist adiabatic lapse rate of about 5 °C/km (Krinner and Genthon, 1999; Minder et al., 2010). The spectra collected
over these sites (Figure 15) show average values of 234.6K for the nunatak and 240.6 K for Inexpressible Island, in line with
their physical temperature, with variations within ±2 K. The microwave emissivity estimated for both sites is then 0.91. The





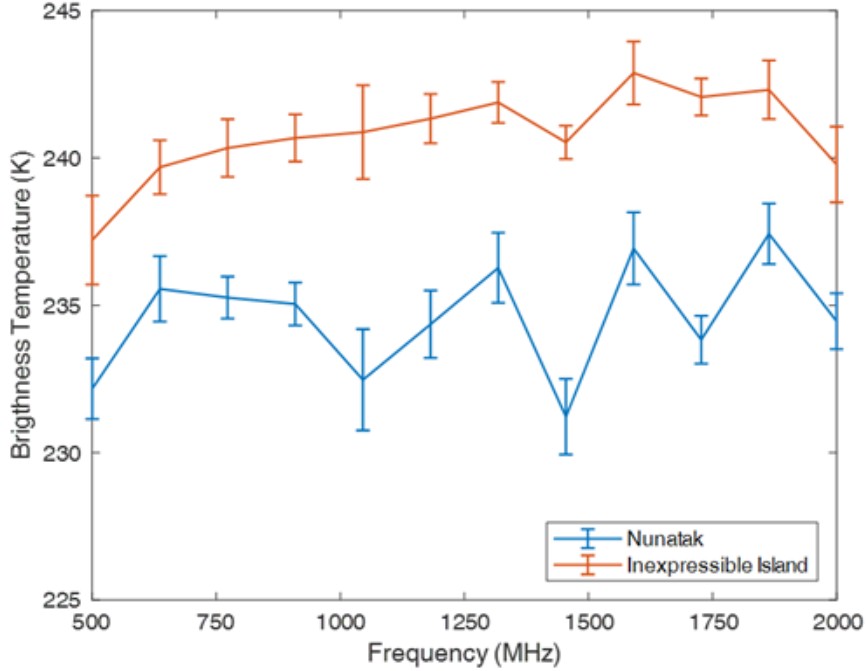

**Figure 15.** Brightness temperature spectra over a nunatak in the David glacier region and over Inexpressible Island. Error bars indicate the range of brightness temperatures observed during the overpass.

similarity of these results to the ~237 K brightness temperature observed for the coastal moraine (centered at 74.730865 °S, 164.001°E) is also suggestive of similar terrain properties in these regions.

## 5.5 Supraglacial lakes

The ability of 500-2000 MHz radiometry to probe the ice subsurface is particularly intriguing for the detection of internal aquifers. A UWBRAD transit over the Northern Foothills from Tethys Bay to the open sea acquired data over several targets including Enigma Lake, a supraglacial lake usually buried by an ice layer whose thickness varies from a few centimeters to about 15 m. The corresponding brightness temperature timeseries on Figure 16 begins over the thick ice of Thetys Bay, then passes over the Strandline Glacier (a small glacier of ~20 m thickness) before observing Enigma Lake followed by Boulder Clay Glacier (a glacier whose thickness reaches a maximum of 120 m) and open sea observations. Brightness temperature spectra from the Boulder Clay and Strandline Glaciers shown in the right portion of Figure 16 decrease with frequency at a rate that apparently depends on the morphological characteristics of the ice (thickness, accumulation, bedrock, etc.). The ice core moraine calibration region is composed by debris of granite and volcanic rocks of about 1 m of thickness, and shows values near the enforced flat spectrum. The spectrum from Enigma Lake shows an increasing trend similar to that for open water but



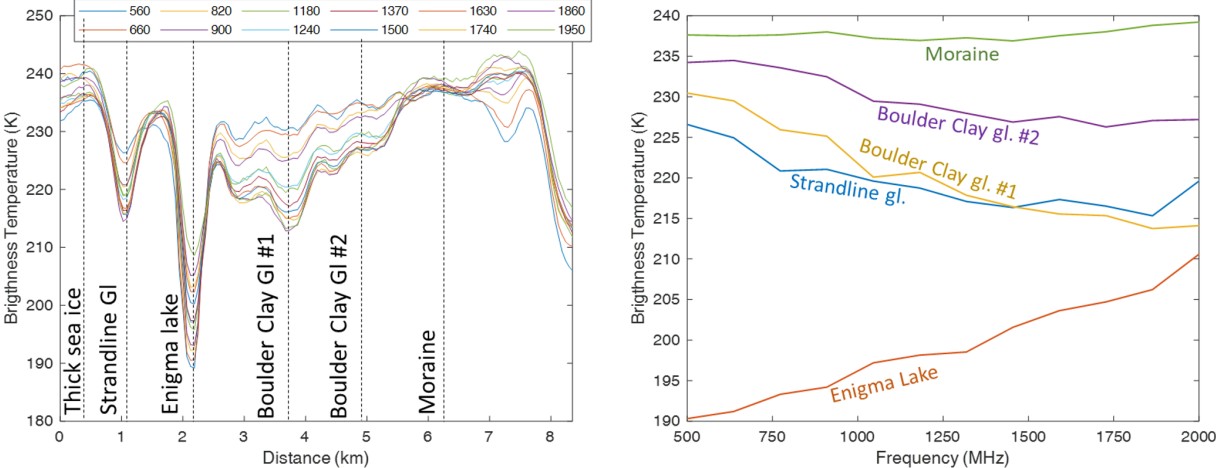

**Figure 16.** (Left) Timeseries of brightness temperature acquired over the Northern Foothills and (Right) spectra of the different targets imaged.

much warmer (from 190 K to 210 K). These distinct signatures for glaciers and the supraglacial lake highlight the potential of ultrawideband radiometry to detect aquifers and other subsurface hydrological features.

# 6  Conclusions

ISSIUMAX was the first campaign in Antarctica aimed at assessing the potential of low-frequency ultrawideband radiometry
for cryospheric studies in coastal and inland areas. The distinct targets observed during the flights (sea ice, land fast sea ice, ice shelf, land ice) and results highlight the benefits of using UWBRAD-like instruments: (i) discriminate between different glacier and sea ice regimes (ii) use of frequencies lower than 1.4 GHz probes the volume of the targets, and (iii) the measurement of a wideband spectrum allows probing different parts of the scene simultaneously. Together these observational characteristics offer a better capability to discriminate surface and/or subsurface properties including for example the presence of snow on sea
ice or ice shelves, the presence of liquid marine water in an ice shelf or brine in sea ice, etc. The principal aim of the campaign was to test the new technique over as many targets as possible in order to assess its potential for cryospheric studies. Although in-situ data were limited for most of the targets, valuable qualitative conclusions nevertheless were achieved, as detailed for each target type in what follows.

The sea ice surveyed in the campaign can be divided into two classes: very thick fast ice (> 2 m) and shallow young ice
(< 50 cm). As expected from theoretical computations (Jezek et al., 2019; Demir et al., 2022), this implies that the brightness temperature spectrum measured was in the first case flat with very high values (240-250 K) and in the second case similar to the values of open water (100-150 K). While we expect that the main benefit of using the entire 0.5-2 GHz band for sea ice is the retrieval of thickness within the 0.5-1.5 m range (Macelloni et al., 2018), the availability of low frequencies (< 1 GHz) and



Tb spectra were shown to improve sea-ice type classification with respect to available present capabilities. For example, Figure
6 right shows that frequencies above 1.25 GHz provide similar information (i.e. the brightness temperature spectrum is flat
above this frequency) while lower frequencies retain sensitivity to distinct ice properties. Also, the availability of multispectral
data with respect to a single L-band channel makes the sensing of properties such as water infiltration (site 8 in Figure 8) or
the presence of snow on-top of the ice (sites 1 and 3 in Figure 6, given that the presence of snow impacts lower frequencies
more significantly than higher ones) possible. While the first result is motivated by the increased penetration depth of lower
frequencies, the second is related to impedance mismatches effect (due to the presence of snow) which are more relevant to
the lower frequencies. Lastly, data collected on thin sea-ice (Figure 10 top right) demonstrated a sensitivity to ice thickness
that increases at low frequencies consistent with the predictions of an electromagnetic model. Although the thickness used as
a reference is subject to error (because it was derived from TIR satellite data), the results suggest that while higher frequencies
saturate at 20/50 cm thickness (depending on salinity), lower frequencies retain sensitivity to higher thickness values. The
comparison of data collected over thin and thick ice also shows the importance of ice salinity in the electromagnetic emission.
Sea ice 20 cm thick with a high brine content (e.g. salinity 12-15 g kg$^{-1}$) can have the same spectral signature as FYI 2 m thick
(salinity 7 g kg$^{-1}$) underscoring the importance of ice salinity information in the thickness retrieval process.

The brightness temperatures collected over glaciers also show the importance of spectral measurements. Over the Priestley
Glacier, the spectral slopes of brightness temperatures are sensitive to the disappearance of the first ∼100 m of firn due to wind
erosion (sites 11 and 12 in Figure 12). At site 11 the target is mainly composed of compacted thick ice and the spectrum slope is
positive (up to 5 K/GHz), while at site 12 the spectrum is more similar to those in the inner part of the ice sheet with a negative
slope of -4 K/GHz (Brogioni et al., 2022). Figure 12 shows distinct spectral features outside the crevasse zones (sites 9, 10 and
11). A deeper analysis of this behavior would require a detailed analysis that includes additional information on the glacier
structure that is beyond the scope of this paper, including consideration of the ice temperature profile and potential impacts of
volume scattering. It is also interesting to note the different spectral behavior before and after the grounding line (e.g. km 25
and km 15 in Figure 12): brightness temperatures decrease at lower frequencies but increase at higher frequencies. This can be
due to the change of the bottom medium (bedrock to sea water) and the structure of the ice at the hinge point. Crevassed areas
are characterized by a marked drop of brightness temperature at all frequencies resulting in a flat spectrum (sites a-e in Figure
12). It is also noted that the same behavior is observed over the Reeves Glacier (blue and orange fluctuations in Figure 13) and
David Glacier (orange fluctuations in Figure 4) although the relative timeseries are not shown here.

The Nansen Ice Shelf was surveyed only in its northern region. Although clear changes in the brightness temperature signal
were noted while transiting from the Reeves Glacier to Inexpressible Island, the snow accumulation zone at the end of the
Priestley Canyon (site 13 in Figure 13) showed particularly distinct impacts. Before and after the snow area, over blue ice, the
spectrum is flat (as observed for crevassed areas of glaciers or for very thick ice) with a value of about 230 K (Figure 14-top).
As the aircraft transited over snow the Tb decreased to an extent that was greater at higher than at lower frequencies. This is in
agreement with electromagnetic scattering theory and suggest the presence of ice lenses and columnar ice not ablated by the
katabatic winds. GPR measurements also support this finding.





Heterogeneous areas with different targets including buried supraglacial lakes were also observed. While transiting from Tethys Bay to Adelie Cove through the Northern Foothills, data were collected over several different scenarios: fast ice,

supraglacial lake, glacier and moraine. As shown in Figure 16, brightness temperatures show fluctuations at all frequencies, but spectral information nevertheless clearly distinguishes these different targets. While the moraine shows a flat spectrum (i.e. the permittivity of rock is almost constant in frequency), snow-covered glaciers have a decreasing spectral trend as seen for the Campbell Glacier Tongue (Figure 6) and the ice sheet (Andrews et al., 2018; Brogioni et al., 2022). Also, Enigma Lake shows an increasing spectral trend which is similar to spectra observed for open water but with a higher brightness temperature level

due to the emission of the overlying ice. It is also noted that, although sea ice and subsurface water both show an increasing spectral trend, sea ice usually shows a saturation (e.g. young ice spectra in Figure 10 top right panel) while the subsurface water spectra show a steadily increasing trend. If needed, this different behavior can used as a discriminator between these two target types.

The campaign also showed that brightness temperature data were observable over the entire 0.5-2 GHz range in Antarctica

after appropriate RFI processing, despite the heavy use of this frequency range by other services in more populated areas.

Overall the ISSIUMAX campaign has demonstrated the capabilities of ultrawideband airborne radiometry for studying near coastal ice sheets that represent a transfer zone between the inland ice sheet and the ocean. The use of brightness temperature spectrum over a wide band allows the discrimination of surface and subsurface characteristics well beyond the current capabilities of existing L-band radiometers. It is noted that some of the interesting features observed within the flights occur at spatial

scales too small to be resolvable from current or planned spaceborne systems, so that only airborne data can be suitable for their monitoring. For example, the width of the outlet glaciers overflown is too small to separate contributions from the ice and from the rocky walls. Nevertheless, the general properties of the different targets observed (e.g. the presence of snow on top of sea ice and ice shelves; the classification of surface types; sea ice thickness) can nevertheless be expected to be observable from space and can contribute to better monitoring of key glaciological parameters and their evolution. The campaign also

showed that further investigations and dedicated experiments that include more extensive in-situ data collection are needed to better understand the complex mechanisms that govern the microwave emission at these frequencies and to extend current electromagnetic models.

*Data availability.* UWBRAD data can be requested to Joel Johnson (johnson.1374@osu.edu) and will be uploaded soon on a public repository. GPR data can be requested to Stefano Urbini (stefano.urbini@ingv.it) and will be uploaded soon on a public repository. Sentinel-1,

Sentinel-2 and Sentinel-3 data are available from the Copernicus Open Access Hub (https://scihub.copernicus.eu, European Space Agency). COSMO-SkyMed© data are not freely available but can be requested to the Italian Space Agency (ASI). ALOS-PALSAR data have been obtained through the ESA Third Party Missions Dissemination Service (https://tpm-ds.eo.esa.int/oads/access/). Landsat-8 are availabe from the USGS Earthexplorer (https://earthexplorer.usgs.gov/). Accumulation Radar and Multichannel Coherent Radar Depth Sounder – MCoRDS datasets are availabe at NSIDC (https://nsidc.org/data/). Quantarctica dataset is available at (https://www.npolar.no/quantarctica/). AWS data

and information were obtained from 'MeteoClimatological Observatory at MZS and Victoria Land' of PNRA' (http://www.climantartide.it).



## Appendix A: UWBRAD calibration and RFI processes

The UWBRAD radiometer includes an internal calibration process that cycles through measurements of the "antenna", "antenna plus noise diode", "reference load", and "reference load plus noise diode" states every 2 seconds. Each measurement corresponds to a 100 ms observation and includes the "fullband" power in each 88 MHz subchannel at 1 ms time resolution,
the "sub-band" power in 512 frequency sub-channels within the 88 MHz subchannel at 1 ms time resolution (called the power spectrogram in what follows), the kurtosis of the fullband power reported at 1 ms time resolution, and the kurtosis in each frequency sub-band at 100 ms time resolution. RFI detection is performed through a combination of algorithms including detectors using the fullband and sub-band kurtosis quantities as well as searches for anomalous results in both time and frequency. Additional information on these algorithms and on the RFI encountered is provided in Andrews et al. (2021). Note that the
exclusion of pixels from subsequent integrations degrades the sensitivity of the integrated product. However, sensitivity can be regained following integration over multiple measurements within the 4.5 s (or longer with degraded spatial resolution) integration time available.

The internal calibration process described in Andrews et al. (2018) is used to compensate for the impact of changes in system temperature or receiver gains on measured data. Internally calibrated measurements are then used to create absolute
calibrated brightness temperatures through an external calibration process that uses observations over the open sea surface and over the coastal moraine region, for which model predictions of the expected brightness temperatures as a function of frequency were applied. Sea surface brightness temperatures were predicted using a two-scale model (Johnson, 2006) applied with wind speed and sea surface temperature information from the European Center for Medium Range Weather Forecasting (ECMWF). An approximate correction for the reflection of cosmic and atmospheric emission was also included in the sea brightness
temperature model prediction. The coastal moraine area (-74.745°S, 164.0271°E) was modeled as having a spectrally-uniform brightness temperature of 237 K which would correspond to a homogeneous interface having a relative permittivity of ~4.5 at the estimated 272 K physical temperature at this location known from an automated weather station (AWS) at this site. This value for the moraine area was assigned based on extensive studies showing the apparent spectral flatness of the observed data at this location as well as the known properties of the terrain in the area. The external calibration process enforces a match
to these predictions as well as agreement in "cross over" measurements through at least squares process that determined the gain and offset parameters in a linear rescaling of the internally calibrated data. The external calibration process was applied first at the level of the 12 x 512 = 6144 individual spectral channels available following initial RFI processing, with additional RFI processing performed after external calibration. The least-squares external calibration algorithm was again applied to the 12 frequency channels obtained after integrating the data over the 512 frequency sub-channels in each channel, and additional
RFI processing was also performed following this combination. An additional step in the external calibration and RFI filtering process as compared to Andrews et al. (2018) enforces the "spectral smoothness" of the observed data, both at the 6144 and 12 channel levels, as previously developed in (Gasiewski et al., 2002). The algorithm performs a 4th order polynomial fit to the measured data versus frequency, and flags outliers from the fit through an iterative process in which the detected outliers are first excluded, a new fit is determined to the remaining data, adjustments to the calibration coefficients of the





excluded channels (for an entire flight) are made based on the current fit in order to attempt to retain these channels, and the process is repeated including the adjusted subchannels and using a refined detection threshold. Following this process, any subchannels showing excessive errors in matching the prescribed brightness temperature values for the sea and moraine calibration targets are ultimately discarded. This approach was required due to the challenges of estimating ultra-wideband brightness temperatures given impedance mismatch effects and a presence of RFI that varies in space and time.

**Appendix B: ISSIUMAX dataset**

Satellite Earth Observation datasets An extensive effort was put in acquiring a wide dataset of microwave and optical satellite products to complement UWBRAD and GPR datasets. In particular we collected:

- 70 SAR SCS 1B products from COSMO-SkyMed® (X-band), 20-27/11/2018, acquired in Stripmap Himage mode, HH pol, 3m ground resolution,

- 118 SAR GRD products from Sentinel-1 (C-band), from 07/01/2018 to 30/11/2018, HH pol, 10m ground resolution,

- 96 SAR GRD products from ALOS-PALSAR (L-band), from 07/10/2007 to 20/01/2011, HH pol, 5m ground resolution,

- SMOS (Kerr et al., 2010) and SMAP (Piepmeier et al., 2016) L-band brightness temperature images for intercalibration with UWBRAD,

- Tb data from JAXA's AMSR-2 at C- to Ka-band to better constrain the inversion algorithms;

- Sentinel-2 optical images, acquired on 22/11/2018, 02/01/2019,

- Landsat-8 optical images collected on 24,27/11/2018,

- ICESaT-2 measurements ATL-7 (DOI = 10.5067/189WL8W8WRH8) and ATL-10 (DOI = 10.5067/189WL8W8WRH8) acquired on 23 and 27/11/2018,

- Medium resolution images from Sentinel-3 and MODIS for monitoring sea ice presence and movements from NASA's
EOSDIS Worldview app (URL = https://worldview.earthdata.nasa.gov/).

- Accumulation Radar (Paden et al., 2014) and Multichannel Coherent Radar Depth Sounder – MCoRDS (Paden et al., 2010) data collected during 2013 Operation Ice Bridge campaign on 19-20/11/2013

- High resolution nadiral L-band Tb over Dome C was also obtained from the past Domecair campaign (Domecair dataset, 2013).

- Ground based L-band brightness temperatures from Domex experiment were used as a reference in UWBRAD calibration (Domex-3 dataset, 2017).





*Author contributions.* MB, MF, JJ, KJ, SU planned the Antarctic campaign. MB, MA, SU, SB performed the Antarctic campaign. MA and JJ processed the UWBRAD dataset. SU processed the GPR dataset. MB and GF processed the satellite dataset. All of the authors contributed to the data analysis. MB prepared the manuscript with contributions from all co-authors.

*Competing interests.* Some authors are members of the editorial board of The Cryosphere. The peer-review process was guided by an independent editor, and the authors have also no other competing interests to declare.

*Acknowledgements.* The ISSIUMAX project was supported by the Italian Antarctic Program – PNRA through the contract 2016/AZ3.02. COSMO-SkyMed© images were provided by the Italian Space Agency under the project id 693. The participation of researchers from The Ohio State University and the University of Michigan were supported by grants from the NASA Cryospheric Science Program. S. Ackley was

supported by NASA Grant #80NSSC19M0194 to UTSA. IFAC activity was partially supported by ASI project Cryorad Follow-On (contract n. 2021-1-U.0). The authors want to thank the MZS operative room and logistics for the help in all the field operations and the KBAL flight crew for the excellent flight executions. Meteorological data and information were obtained from 'MeteoClimatological Observatory at MZS and Victoria Land' of PNRA - http://www.climantartide.it. The scientific results and conclusions, as well as any views or opinions expressed herein, are those of the authors and do not necessarily reflect those of NOAA or the U.S. Department of Commerce.



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
