# Peer review of "Ice Sheet and Sea Ice Ultrawideband Microwave radiometric Airborne eXperiment (ISSIUMAX) in Antarctica: first results from Terra Nova Bay"

_The Cryosphere, 2022_

## Referee Comment (RC1)

**Review for:**

"ICE SHEET AND SEA ICE ULTRAWIDEBAND MICROWAVE RADIOMETRIC AIRBORNE EXPERIMENT (ISSIUMAX) IN ANTARCTICA: FIRST RESULTS FROM TERRA NOVA BAY"

by Marco Brogioni *et al.*

submitted to *The Cryosphere Discussion*

Manuscript ID: TC-2022-59

**Synopsis:**

In this paper, the first airborne microwave wide-band radiometer observation over Antarctica is explained. It is very well written with great results of the of the campaign. I believe this manuscript is applicable for publication in this journal after addressing few comments (some editorial).

**Comments to the Author:**

1. The two top figures in Fig. 1 are very close to each other and misleading for readers' eyes. Can you please separate them somehow? Perhaps tag the figures as a, b, c, d. Or add colorbar to the left figure, or make the box around the figure to be bolder and more distinguishable.
2. How much loss of data due to an RFI can be tolerated? Have you looked into this?
3. Regarding the claim in line 5 (in the abstract) where it is mentioned that lower frequency shows warmer since it senses deeper into the ice where it is physically warmer, I think this statement is not necessarily true, and any boundary with a huge dielectric constant difference at deeper layers can drastically drop the L-band or lower frequency TB. Can you mention this point more cautiously, or explain it further?
4. Regarding my third comment, lower right figure in Fig .1, why lower frequency of 540 MHz is the coldest before about 21:13 UTC, then it becomes hottest after that? This is opposite of the claim you had for lower frequency. Can you please explain this?
5. In line 260, I parenthesis after nearly nd) there is a missing 2 I believe. Please correct this.
6. Please explain for the reader that why the RTM model in bottom right figure did not go less than about 4cm.
7. In figure 14, what is the reason of short drop of TB at around 25 km? Is there any ice lens? Same thing happened between 20 and 25 km also.

---

## Author Comment (AC1)

The authors want to thank the reviewer for his comments and suggestions that helped improving the paper. Our answers and comments have been written in magenta.

REVIEWER #1

The paper is very well written. I only have few comments to be addressed before it can get published. Pleases see attached. Thanks.

Synopsis:
In this paper, the first airborne microwave wide-band radiometer observation over Antarctica is explained. It is very well written with great results of the of the campaign. I believe this manuscript is applicable for publication in this journal after addressing few comments (some editorial).

We appreciate the general comment of the anonymous reviewer. We did our best to expose in the clearest way the potential of this new approach to microwave radiometry in the very low-frequency range, as well describe the first results in Antarctica over such a variety of scenarios.

Comments to the Author:
1. The two top figures in Fig. 1 are very close to each other and misleading for readers' eyes. Can you please separate them somehow? Perhaps tag the figures as a, b, c, d. Or add colorbar to the left figure, or make the box around the figure to be bolder and more distinguishable.

Thanks for the suggestion. The figure has been improved resulting in a higher clarity. It is reported hereinafter for your convenience.

[Figure]

Figure 1. (top panel) 6144 frequency channel spectrograms of brightness temperatures (Kelvin) before (a) and after (b) application of the iterative RFI filtering and external calibration procedure. (c) Photograph of UWBRAD antenna periscope and equipment rack aboard Twin Otter aircraft on Ken Borek Airlines. (d) Twelve channel integrated brightness temperatures corresponding to the spectrograms shown in the upper plots. Labels indicate type of targets observed along the transect.

2. How much loss of data due to an RFI can be tolerated? Have you looked into this?

This is an important point that has been discussed in past works. The answer can be complex.
From a scientific point of view the spectrum measured on a natural target in this frequency range is usually monotonic (at least for 100 MHz intervals) so that losing one or more channels doesn't impact the data analysis significantly, especially if the channels lost are not consecutive. In general, the loss of an entire channel in the higher end of the range is likely to have a smaller impact than loss of a lower frequency channel given that the penetration depth tends to follow an exponential law.
From an engineering perspective we calculated that the loss of up to 75% of the data for an acquisition will cause the NEdT to double from 0.5K to 1K, a value that is still usable for the data analysis and retrieval. This sentence will be added in our revised manuscript at the end of line 497.

In the paper by Andrews et al. 2021, 10.1109/TGRS.2021.3090945, we tried to assess the impact of RFI on the Tb measurements in detail. In particular, in section IV of this reference, the data collected during both Arctic and Antarctic campaigns were analyzed and the statistics of the data loss produced. Figures 10 and 11 of this reference compare the data lost due to RFI with respect to the ideal case (UWBRAD observing liquid nitrogen, so no RFI input). The plots show the percentage of time lost for a given channel. e.g. for LN2 observations, 15% of the data is always lost due to the RFI processor settings. However the loss of 100% of the channel is extremely rare (less than 0.001% of the time). It is interesting to note that in Antarctica the data loss is very small, independently of the presence of human infrastructures. Panel (g) of Fig 10 in Andrews et al 2022 shows the percentage of acquisitions in which the data are completely lost. It can happen that over specific targets some channels are lost but this is limited to very specific bands and only for a limited percentage of the time.

In summary, at present the impact of RFI on the Tb measurements seems to be modest especially in polar regions. We add the following sentences to Appendix A:

line 495 "…integrated product. It has been estimated that the loss of up to 75% of the data for an acquisition will cause the NEdT to double from 0.5K to 1K in that specific channel. However…"

line 497 "…time available. A detailed assessment of the RFI impact on the UWBRAD measurements was performed by Andrews et al. (2021) by using experimental data collected in both polar regions. As expected, RFI have a strong impact on data acquisition over areas having higher human presence (e.g. Canada) but lesser (Greenland) to negligible (Antarctica) in remote regions. In Greenland the loss of the 75% of the acquisitions happened less than the 10% of the time and only for a few specific channels."

3. Regarding the claim in line 5 (in the abstract) where it is mentioned that lower frequency shows warmer since it senses deeper into the ice where it is physically warmer, I think this statement is not necessarily true, and any boundary with a huge dielectric constant difference at deeper layers can drastically drop the L-band or lower frequency TB. Can you mention this point more cautiously, or explain it further?

The sentence is only intended to provide the reader with a rule-of-thumb for interpreting the data we will show. It is true that any strong dielectric discontinuity affects the electromagnetic emission and deviates by the general behavior described for inland ice (in particular on the East Antarctic Plateau). This is exactly what we notice over the Nansen ice shelf, where the subsurface structures created in the firn by the seasonal melt/refreeze "mask" the emission from the ice below. Also, it is the basis for subsurface aquifers monitoring, a topic we are working on especially for Greenland and at which we hint in section 5.5 while describing the data over Enigma Lake.

The sentence "Generally, the brightness temperatures over the inland ice sheet were warmest at the lowest frequencies consistent with models that predict that those channels sensed the deeper, warmer parts of the ice sheet." has been reworded as:

"Generally, the brightness temperatures over a vertically homogeneous ice sheet are warmest at the lowest frequencies consistent with models that predict that those channels sensed the deeper, warmer parts of the ice sheet. Vertical heterogeneities in the ice property profiles can alter this basic interpretation of the signal."

4. Regarding my third comment, lower right figure in Fig .1, why lower frequency of 540 MHz is the coldest before about 21:13 UTC, then it becomes hottest after that? This is opposite of the claim you had for lower frequency. Can you please explain this?

The previous statement refers to inland ice sheets while Fig.1 illustrates, as an example, the spectral behavior over different components of the cryosphere.  Over open water, all channels are cool with the lowest channels coldest because of the dielectric behavior of the water.  Over thinner sea ice (as at 21:13), the low frequency are more sensitive to the radiometrically cold ocean water underlying the ice.  Over the thick, low-loss glacier ice of the Campbell Glacier, the low frequencies are more sensitive to the warming thermal gradient in the ice. The figure was intended to provide the reader an idea of how the timeseries collected by UWBRAD looked when it observed different natural targets. In particular, the data before 21:13 UTC refers to open water and grey ice, which explains the low Tb values. Soon after moraine and fast ice are observed with completely different spectral features. Comment 3 refers to the inner ice sheet which was not covered in figure 1.

Thanks to your comments we recognize that figure can be improved in order to help the reader understand the general behavior of Tb. We have annotated Fig. 1 as

5. In line 260, I parenthesis after nearly nd) there is a missing 2 I believe. Please correct this.
Done. Thanks for noticing.

6. Please explain for the reader that why the RTM model in bottom right figure did not go less than about 4cm.

We choose to show the RTM simulations starting from 5 cm because RTM is a widely used incoherent model that assumes a single homogeneous planar slab of sea ice above water.  The model breaks down when the overlying layer becomes thinner than a few cm of sea ice.  More specifically, a layer thinner than a quarter of the wavelength (15 cm at 0.5 GHz and 3.5 cm at 2 GHz) cannot be treated with conventional/simple wave propagation approach and a coherent RT should

be used. As described in the text (line 297) the analysis was intended to show the potential sensitivity of UWB radiometry to SIT even for high saline shallow ice and corroborate it with electromagnetic model simulations. At line 300 we added the following sentence in order to clarify such limitation:

"RTM simulations assume a finite thickness slab of sea ice over water, and the calculation starts from a minimum thickness of 5 cm since for lower values the incoherent model fails to simulate correctly the e.m. emission. Actually, layers thinner than a quarter of the wavelength (about 15 cm at 0.5 GHz and 3.5 cm at 2 GHz) cannot be treated with conventional/simple wave propagation approach and a coherent RT should be used (which requires a highly accurate characterization of the medium surfaces seldom available in such scenarios)."

Hereinafter we report the same plots with RTM simulations starting at 3 cm but we believe that it doesn't add too much information to the paper.

[Figure]

7. In figure 14, what is the reason of short drop of TB at around 25 km? Is there any ice lens? Same thing happened between 20 and 25 km also.

We thank the reviewer for the accurate observation. We checked the full ISSIUMAX EO dataset (optical and SAR images, microwave sounder echograms, etc.) and we didn't find any significant cause for the Tb fluctuations. Then we moved to check the UWBRAD raw data and we found that the Tb glitches were caused by electromagnetic interferences affecting the entire RF chain and the relative internal calibration. This made possible the RFI processor to not detect the contamination. We deleted the data from the figure and edited Fig. 14 label as:
"Figure 14. Comparison between UWBRAD and SAR data collected across the snow zone on the Nansen Ice Shelf (top) and GPR acquisitions (bottom). Red colorscale indicates the intensity of GPR

radargram. Shaded area indicates data collected over site 13. Data missing around km 25 was caused but a strong electromagnetic interference."

At the same time we checked the entire UWBRAD dataset too for excluding similar issues on other sites. No additional issues were found and Fig. 14 resulted the only one affected by this problem.